# First Known Report of *mcr*-Harboring Enterobacteriaceae in the Dominican Republic

**DOI:** 10.3390/ijerph20065123

**Published:** 2023-03-14

**Authors:** Angela Perdomo, Hattie E. Webb, Marie Bugarel, Cindy R. Friedman, Louise K. Francois Watkins, Guy H. Loneragan, Alexandra Calle

**Affiliations:** 1School of Veterinary Medicine, Texas Tech University, Amarillo, TX 79409, USA; 2Division of Foodborne, Waterborne, and Environmental Diseases, Centers for Disease Control and Prevention, Atlanta, GA 30329, USA; 3Division of Research and Development Resources, BioMérieux, 69795 Lyon, France; 4Division of Global Migration and Quarantine, Centers for Disease Control and Prevention, Atlanta, GA 30329, USA

**Keywords:** antimicrobial resistance, colistin resistance, Enterobacteriaceae, *Escherichia coli*, IncX4, *mcr* genes

## Abstract

Colistin is a last-resort antibiotic used to treat infections caused by multidrug-resistant Gram-negative bacteria. People with a history of travel to the Dominican Republic have become sick with pathogenic bacteria carrying the mobile colistin resistance gene, *mcr*-1, during and after traveling. This investigation aimed to identify *mcr* genes in Enterobacteriaceae isolated from food animal sources in the Dominican Republic. Three hundred and eleven samples were tested, from which 1354 bacterial isolates were obtained. Real-time PCR tests showed that 70.7% (220 out of 311) of the samples and 3.2% (44 out of 1354) of the isolates tested positive for the *mcr* gene. All RT-PCR presumptive *mcr*-positive isolates (n = 44) and a subset (n = 133) of RT-PCR presumptive *mcr*-negative isolates were subjected to whole-genome sequencing. WGS analysis showed that 39 isolates carried the *mcr* gene, with 37 confirmed as positive through RT-PCR and two as negative. Further, all of the *mcr*-positive genomes were identified as *Escherichia coli* and all contained a IncX4 plasmid replicon. Resistant determinants for other antibiotics important for human health were found in almost all isolates carrying *mcr* genes.

## 1. Introduction

Antibiotic resistance is a growing threat to global public health and is considered a critical concern within the One Health framework. The One Health approach interconnections describe the emergence, evolution, and spread of antibiotic-resistant microorganisms on both a local and global scale, which poses a significant risk to public health [1]. The overuse and misuse of antibiotics in human and veterinary medicine contribute to the selection and dissemination of antibiotic-resistance among bacteria leading to difficult-to-treat infections that negatively impact human and animal health, as well as the productivity of animals [2,3]. Aside from the overuse of antimicrobials, the utilization of colistin and other critically important antibiotics in food-producing animals also represents a big threat to healthcare systems [4]. Colistin is an old polymyxin considered a crucial last-resort antibiotic used to treat infections caused by multidrug-resistant (MDR) Gram-negative bacteria. The first plasmid-mediated colistin resistance mechanism in Enterobacteriaceae, *mcr*-1, was reported in China in 2015 [5]. Since then, retrospective and prospective screening efforts have globally detected *mcr*-1 in Enterobacteriaceae species collected from humans, animals, food, and the environment [6,7]. Although *mcr*-1 is the most observed *mcr* gene, additional *mcr* genes (from *mcr*-2 to *mcr*-10) and their variations have been described [8,9].

In the United States, where colistin use has never been approved in livestock, the first *mcr* gene reported was in an *mcr*-1-positive *Escherichia coli* isolate collected in 2016 from a urine sample from a woman in Pennsylvania. In the same year, the United States Department of Agriculture (USDA) reported *mcr*-1 in bacteria isolated from the intestinal tract of two pigs [10,11]. Continued retrospective and prospective screening in the U.S. revealed that most *mcr* genes are rarely detected in the routine surveillance of four major foodborne bacteria (*Salmonella*, *Campylobacter*, *Escherichia coli*, and *Enterococcus*) from humans, animals at slaughter, and retail meats. Such surveillance efforts have also linked *mcr*-positive clinical isolates and international travel within seven days before illness onset. From 2016 to 2021, 14 *mcr*-positive *Salmonella* were isolated from U.S. patients who traveled to the Dominican Republic (DR) [12,13]. Those data suggest that U.S. travelers may be exposed to and acquire enteric pathogens carrying *mcr* genes abroad by consuming contaminated food or water. This work aimed to identify *mcr* genes in Enterobacteriaceae isolated from animals, food, and animal environments from the DR to understand potential sources of human exposure better. The specific objectives were to: (i) estimate sample-level prevalence by identifying *mcr* genes directly from enriched samples; (ii) estimate isolate-level prevalence by detecting *mcr* genes in bacterial isolates obtained from the samples; (iii) confirm taxonomic group, specific antimicrobial resistant determinants, and plasmid replicon types for RT-PCR presumptive *mcr*-positive isolates using whole genome sequencing (WGS).

## 2. Materials and Methods

### 2.1. Sample Collection

Samples were collected from raw meat (beef, pork, and broiler chickens [poultry]), feces from the corresponding food-producing animals, water for animal consumption, and animal feed. These samples were obtained from different locations around the country, such as Bayaguana, Hato Mayor, Higuey, La Romana, La Vega, Mao, Moca, Puerto Plata, San Francisco de Macoris, San Pedro de Macoris, Santiago de los Caballeros, Santo Domingo, and Santo Domingo Norte (Figure 1). Sampling sites and required authorizations to visit the farms were procured with the help of the Ministry of Agriculture in the DR. Meat samples were purchased from supermarkets and small butcher shops, placed into sterile bags, and transported the same day under refrigeration for further testing. Fecal samples from beef cattle and swine consisted of rectal specimens collected by government veterinarians using sterile palpation gloves and placed into sterile cups. Poultry litter was sampled using sterile pre-moistened boot swabs (Envirobootie™, Hardy Diagnostics). Two individuals wore boot swabs and walked around the poultry pen for 5 min. Each boot swab (left and right foot) was placed into separate sterile bags and treated as an independent sample. Water samples were obtained from animal waterers from farming operation facilities (beef cattle, swine, or poultry) and placed into sterile cups. Animal feed was obtained either from open containers at the farms or warehouses supplying pig farms and poultry houses. Beef cattle feed samples were limited to only one farm. All samples were labeled accordingly (date, type of sample, animal origin, location, and name of farm/market/shop) and transported the same day in coolers to the official testing microbiology laboratory affiliated with the Ministry of Agriculture in Santo Domingo.

### 2.2. Sample Preparation and Enrichment

Buffered peptone water (BPW, MilliporeSigma, Burlington, MA, USA) was supplemented with 1 mg/L of colistin (BPW1) and BPW without colistin (BPW0) was prepared following the manufacturer’s instructions. Glass bottles were used since it is known that colistin easily adheres to plastic [14]. All samples were separately enriched with 90 mL BPW0 and BPW1. Meat cuts and boot swabs were rinsed with 250 mL of BPW, and 10 mL of the rinse was transferred into each enrichment solution. For fecal and feed samples, 10 g were used for enrichment. For water samples, 10 mL were transferred to each enrichment bottle. Samples were incubated at 37 °C for 24 h.

### 2.3. Enterobacteriaceae Detection and Isolation

Following incubation, samples were cultured on violet red bile glucose agar plates (VRBG, BD Difco^®^, Sparks, MD, USA) supplemented with colistin (1 mg/L [VRBG1] and 0 mg/L [VRBG0]) and incubated at 37 °C for 24 h. Presumptive Enterobacteriaceae colonies, selected based on typical morphology, were cultured on tryptic soy agar (TSA, MilliporeSigma, Burlington, MA, USA), supplemented with colistin (1 mg/L [TSA1] and 0 mg/L [TSA0]), and incubated at 37 °C for 24 h. Colonies were transferred to TSA slants supplemented with colistin and stored at 4 °C for analysis.

### 2.4. Salmonella Detection and Isolation

Samples were pre-enriched in BPW0 and incubated at 37 °C for 24 h. A subsequent selective enrichment was conducted by transferring 1 mL into 9 mL of Rappaport–Vassiliadis broth (RV, BD Difco^®^, Sparks, MD, USA) and 1 mL into 9 mL of tetrathionate broth (TT, BD Difco^®^, Sparks, MD, USA), followed by incubation at 42 °C for 24 h. Following incubation, each selective enrichment, was streaked onto Xylose Lysine Tergitol 4 agar plates (XLT4, BD Difco^®^, Sparks, MD, USA) and brilliant green sulfa agar (BGS, BD Difco^®^, Sparks, MD, USA), and incubated at 37 °C for 24 h. *Salmonella* was isolated following steps similar to Enterobacteriaceae, including culture on TSA and transferred to new TSA slants.

### 2.5. DNA Extraction from Samples

DNA was extracted from the BPW enrichments not supplemented with colistin. QIAamp PowerFecal DNA Kit (QIAGEN^®^ Group, Germantown, MD, USA) was used for fecal samples, and the InstaGene matrix kit (Bio-Rad Laboratories, Hercules, CA, USA) for feed, water, and meat cut samples. The manufacturer’s protocols were followed in both cases.

### 2.6. DNA Extraction from the Isolates

Presumptive *Salmonella* isolates were streaked into a new TSA plate without colistin and incubated overnight at 37 °C. Colonies were selected and inoculated in Luria broth (LB, Remel, Lenexa, KS, USA) and incubated at 37 °C for 18 h in an incubator shaker. The presumptive Enterobacteriaceae isolates were transferred to VRBG supplemented with 1 mg/L and 0 mg/L of colistin and incubated at 37 °C for 24 h. Following incubation, one colony was inoculated in TSB supplemented with the corresponding colistin concentrations and incubated at 37 °C for 18 h in an incubator shaker. An aliquot of 150 mL from the overnight culture was used to extract genomic DNA using a boiling method. Samples were centrifuged at 4500 rpm for 3 min at 4 °C, and the pellet was resuspended in 30 mL of molecular-grade water and centrifuged again. Samples were stored at −20 °C until further analysis. Separately, 850 µL of the overnight TSB enrichment was preserved with 150 µL of 80% glycerol at −80 °C for in-house culture collection.

### 2.7. Multiplex RT-PCR for mcr Gene Detection

Multiplex real-time polymerase chain reaction (RT-PCR) was conducted to detect *mcr*-1 to *mcr*-8 genes. The multiplex RT-PCR assay was designed based on reference sequences that are accessible in the ResFinder database by the time of the study. It included 14 variants of *mcr*1, two variants of *mcr*2, 25 variants of *mcr*3, six variants of *mcr*4, two variants of *mcr*5, one variant of *mcr*6, one variant of *mcr*7, and one variant of *mcr*8. The multiplex RT-PCR was optimized as follows: 25 mL PCR reactions including 1× Brilliant II QPCR Master Mix (Agilent, Santa Clara, CA, USA), *mcr*-3, *mcr*-4, *mcr*-5, *mcr*-7, and *mcr*-8 forward and reverse primers at the final concentration of 0.6 µM, *mcr*126 forward and reverse primers at 0.3 µM, *mcr*126 probe at 0.7 µM, and *mcr*58 and *mcr*347 probes at 0.8 µM and a DNA input of 3 mL. The RT-PCR thermal program was set-up as follows: an initial denaturation of 10 min at 95 °C, followed by 35 cycles, including denaturation for 30 s at 95 °C and an annealing/elongation step of 60 s at 66 °C. Primers and probes were designed and labelled as shown in Table 1.

Positive controls for *mcr*-1 and *mcr*-5 were obtained using *Salmonella* reference strains 12CEB2196SAL [15] and S12LNR3592 [16], respectively. These strains belong to the French reference laboratory (Anses, Maisons-Alfort, France). Positive controls for *mcr*-2, *mcr*-3, *mcr*-4, *mcr*-6, *mcr*-7, and *mcr*-8 were obtained by electroporating *Salmonella* 10TTU468x (obtained at Texas Tech University [TTU] Food Microbiology Laboratory culture collection) with AmpR plasmids, cloned by the manufacturer (IDT, Newark, NJ, USA) with *mcr*-2.1, *mcr*-3.1, *mcr*-4.1, *mcr*-6.1, *mcr*-7.1, or *mcr*-8.1 sequences obtained from the ResFinder database. The *mcr*-9 and *mcr*-10 genes were not included since they were not described when this study was designed.

### 2.8. Whole Genome Sequencing

All RT-PCR presumptive *mcr*-positive isolates were subjected to whole genome sequencing (WGS). A subset of RT-PCR presumptive *mcr*-negative isolates (n = 133) was selected randomly (using the RAND Microsoft Excel function with weight) for sequencing. DNA extraction was performed using GenEluteTM bacterial genomic DNA kit (Sigma-Aldrich, NA2100, NA2110, or NA2120, St Louis, MO, USA), following the manufacturer’s protocol. Libraries were constructed with 5 µL genomic DNA (~100–200 ng) and Nextera DNA Flex Library Prep Kit (Illumina) and quantified on the fluorometer using PicoGreen, according to the manufacturer’s protocol. The pool was analyzed using an Illumina MiSeq Reagent Nano kit v2 (300 cycles) on Illumina MiSeq. Then, 10 µL of the pooled library at a final concentration of 200 pM was sequenced using an Illumina 250 × 250 NovaSeq SP Flow Cell (500 cycles) on the Illumina NovaSeq-6000 sequencing facility (TTU Center for Genetics and Biotechnology, Lubbock, TX, USA). Genotypic characterization was performed on all 44 RT-PCR presumptive *mcr*-positive isolates. From the randomly selected 133 RT-PCR presumptive *mcr*-negative isolates, 20 were discarded due to low coverage during the short read sequencing. Therefore, plasmid characterization and antimicrobial resistance (AMR) profiling were conducted on 113 isolates. Genus and species identification were performed using FastANI (https://github.com/ParBLiSS/FastANI) and the Genome Taxonomy database (https://data.ace.uq.edu.au/public/gtdb/data/releases/release95/; accessed on 16 July 2020). De novo assemblies were generated using shovill v1.0.9 (https://github.com/tseemann/shovill), and contigs with coverage below 10% average genome coverage were excluded. Staramr v0.4.0 (https://github.com/phac-nml/staramr) was used to screen assemblies for resistance determinants using the ResFinder database (Center for Genomic Epidemiology [CGE], https://cge.cbs.dtu.dk; downloaded 30 July 2020, 90% identity, 50% cutoff) and CGE PointFinder scheme for *Salmonella* spp. ARIBA v2.12.0 and the CGE PointFinder database were used to screen *E. coli* for point mutations. Plasmid replicons were identified using abricate v0.8.10 and a database adapted from CGE PlasmidFinder (90% identity, 60% cutoff). Multilocus sequence type was determined from de novo assemblies using the Tseemann MLST tool (https://github.com/tseemann/mlst) and the PubMLST database [17].

## 3. Results

In total, 311 samples were collected from 17 farms, 21 supermarkets, and 12 butcher shops in the DR. Table 2 shows the detail of each type of sample collected and the proportion of RT-PCR presumptive *mcr*-positive samples by animal source and sample type. Based on the multiplex RT-PCR, 70.7% (n = 220) of the samples collected carried one or more *mcr* genes. The proportion of RT-PCR presumptive *mcr*-positive samples collected from beef cattle, swine, and poultry was 58.1% (54/93), 74.3% (81/109), and 78.0% (85/109), respectively. The overall proportion of RT-PCR presumptive *mcr*-positive samples based on sample type for feces, feed, water, and meat, was 79.1% (91/115), 57.7% (15/26), 60.0% (9/15), and 67.7% (105/155), respectively.

### 3.1. mcr Prevalence at the Isolate Level

A total of 1354 isolates were recovered from the 311 samples. Multiplex RT-PCR showed that 3.2% (n = 44) of the isolates carried *mcr* genes. Of the 44 presumptive *mcr*-positive isolates, 81.8% (n = 36) were obtained from swine sources, followed by 11.4% (n = 5) from beef cattle and 6.8% (n = 3) from poultry (Table 3). When grouping isolates by sample type, the highest *mcr* proportion was observed among isolates recovered from fecal samples at 7.9% (n = 38), followed by feed, and meat samples (4.8% [n = 3] and 0.4% [n = 3]). Multiple isolates were collected from each sample; thus, closely related or identical strains may be represented more than once.

### 3.2. Whole Genome Sequencing Analysis for mcr Confirmation

Presumptive RT-PCR *mcr*-positive (n = 44) and *mcr*-negative (n = 133) isolates were subjected to WGS analysis. Of the 44 RT-PCR presumptive *mcr*-positive isolates, 84.1% (n = 37) were confirmed as *mcr*-positive (Appendix A). WGS also revealed that two RT-PCR presumptive *mcr*-negative isolates carried *mcr* genes, and both were identified as *E. coli* (*mcr*-1-positive *E. coli* [DR2-004A1 and DR2-007A1]). Altogether, the 39 *mcr*-1-positive *E. coli* isolates included the following sequence types: ST10, ST29, ST48, ST50, ST191, ST410, ST871, ST1602, ST1771, ST6778, and ST8233 (Appendix A). From the 133 *mcr*-negative randomly selected isolates subjected to WGS, 113 sequences were included in our analysis (20 sequences did not have the sufficient coverage and were thus excluded from the analysis). Sequences were classified as *Morganella morganii* 30.1% (n = 40), *Salmonella enterica* 24.1% (n = 32), *E. coli* 8.3% (n = 11), *Proteus* spp 6.8% (n = 9), *Providencia* spp 5.3% (n = 7), *Hafnia* spp 3.4% (n = 5), *Citrobacter* spp 3.0% (n = 4), *Enterobacter* spp 1.5% (n = 2), *Shewanella algae* 1.5% (n = 2), and *Moellerella wisconsensis* 0.7% (n = 1). Of the 39 *mcr*-positive isolates, 92.3% (n = 36) originated from swine feces, 5.1% (n = 2) from poultry feed, and 2.6% (n = 1) from beef (meat) (Appendix A). These 39 *mcr*-positive *E. coli* isolates were recovered from 17 out of the 311 samples. Furthermore, these isolates were obtained from samples collected in either the Santo Domingo Province (Santo Domingo and Santo Domingo Norte; n = 8) or North-Central DR (La Vega [n = 7] and Moca [n = 2]).

The *mcr*-positive isolates, including those that mediate resistance harbored many additional antimicrobial resistance determinants to aminoglycosides, quinolones, beta-lactams, macrolides, tetracyclines, phenicol, and/or folate pathway inhibitors (including trimethoprim and sulfonamides (Figure 2). Of the 39 isolates carrying *mcr* genes, 38 harbored additional antimicrobial-resistance determinants and all 38 were predicted to be MDR (resistant to three or more antimicrobial classes). Moreover, many of these determinants mediate resistance to additional critically important antimicrobials for human medicine, including aminoglycosides, quinolones, macrolides, and subclasses of the beta-lactam class. The aminoglycoside modifying enzymes *aph(3″)-Ib* (87.2%) and *aph(6)-Id* (84.6%) were commonly present. The quinolone resistance determining region (QRDR) mutation in *gyrA(S83L)* was present in 51.3% (n = 20), *gyrA(D87N)* in 28.2% (n = 11), *gyrA(D87G)* in 2.6% (n = 1), and *parC(S80I)* in 33.3% (n = 13) of the isolates. Moreover, the *qnrB19* and *qnrS1* genes, which confer reduced susceptibility to quinolones in Enterobacteriaceae, were observed in 61.5% (n = 24) and 2.6% (n = 1), respectively. The *bla*_TEM_ gene variants, especially *bla*_TEM-1B_, were present in 43.6% (n = 17). Furthermore, the *mef(B)* macrolide-efflux pump gene was found in 25.6% (n = 10) of the isolates, followed by the presence of both *mph(A)* and *erm(42)* genes in 30.8% (n = 12). Among the *mcr*-positive isolates, the most predominant plasmid replicons found were IncX4 (100.0%), ColE1 (61.5%), Col(pHAD28) (61.5%), and IncR (56.4%) (Figure 3). Other replicon types known for carrying *mcr* genes were less common among the *mcr*-positive isolates and included IncX1 (48.7%), IncFIB (46.2%), IncFII (23.1%), and IncHI2 (2.6%).

As previously noted, all 39 *mcr*-1-positive isolates harbored an IncX4 plasmid replicon. In most *mcr*-1-positive isolates (35/39), both *mcr*-1 and IncX4 were on the same contig However, six different contig variations were observed (Figure 4A–F). The most common (Figure 4B, n = 23), harbored a partial IS*26* with direct repeats (TCACACAG) at each end of the contig. Contig variations C–F (Figure 4) also contained both *mcr*-1 and IncX4 on the same contig; however, they differed by the insertion sequences (IS*1294*, IS*903C*, or IS*10*) at one end resulting in shorter contigs. When *mcr*-1 and the IncX4 replicon were found on different contigs (Figure 4G, n = 4), portions of IS*3* with direct repeats (AGT) were found on one end of each contig, indicating these contigs may be linked (Figure 4 G). In three of the 39, a single partial (1023/1070 bp) insertion sequence, IS*Apl1*, was found 2428 bp upstream of the *mcr*-1 (Figure 4A). No evidence of an IS*Apl1* was identified adjacent to *mcr*-1 in the remaining 36 isolates.

## 4. Discussion

The dissemination of MDR Gram-negative bacteria carrying genes that confer resistance to critically important antibiotics through the food chain represents a global health threat. Mobile colistin resistance genes have been detected in Gram-negative bacteria around the world. In many geographic regions, the prevalence of *mcr* genes remains uncharacterized or underreported due to the lack of systematic research on this topic. Strengthening knowledge related to the prevalence and distribution patterns of resistance determinants, such as *mcr*, among pathogens and geographical regions can be foundational to inform action against their selection and dissemination. The U.S. has national surveillance efforts in place, and it is known that *mcr* genes are rare in the country. When *mcr* genes have been observed, they are often associated with U.S. patients that report recent travel, often to the DR [12]. The mobility of humans through global travel is a key factor in disseminating antibiotic-resistant organisms. Travelers are exposed to bacteria through food or water and could subsequently acquire and spread *mcr*-positive Enterobacteriaceae. This study is the first known report of the crude prevalence of *mcr* genes using RT-PCR at the sample, and bacterial isolate levels obtained from food-producing animals and environments in the DR. Whole genome sequencing was used to confirm the presence of *mcr* genes in the recovered *E. coli* isolates.

Our RT-PCR results indicate that *mcr* genes were not uncommon in samples from beef cattle, swine, poultry and their environments in the DR. A higher prevalence of *mcr* genes was observed in swine compared with beef cattle and poultry. Whole genome sequencing data also indicated that *mcr*-1 was not uncommon in *E. coli* from swine in the DR, particularly in the feces. Other researchers have reported plasmid-mediated colistin-resistant genes in swine from other geographic regions. For example, a study in China found that the prevalence of *mcr* genes at the sample level was higher in pigs than in other animals, such as poultry. The high level was associated with the prolonged and widespread use of colistin as a growth promoter in animals in China [18,19,20]. Another study on pig farms in Portugal reported a high prevalence of the *mcr*-1 gene in *E. coli* isolates [21]. Researchers have attributed the frequency of *mcr* genes in swine to the use of colistin sulfate for growth promotion or to control infections caused by *E. coli* in pigs, such as post-weaning diarrhea and edema disease [5,22,23]. In the DR, 36 colistin-containing products are approved for veterinary use in poultry, cattle, goats, sheep, pigs, cats, or dogs by the Ministry of Agriculture [24]. However, there are no known official animal antibiotic regulations, and antibiotic use and sales data still need to be included. During the sample collection of this project, we observed farmers adding colistin to the animal feed on swine farms but not on other animal holdings.

Interestingly, only one *mcr* gene variant (*mcr*-1) was found in this study—in one organism, *E. coli*—despite using targeted isolation methods to identify eight variants of *mcr* genes among Enterobacteriaceae more generally. Despite reports from many countries of *Salmonella* spp. in food, animals, and clinical specimens carrying *mcr* genes, and reports of *mcr*-positive *Salmonella* spp. isolated from U.S. travelers returning from the DR [12], *mcr* genes were not detected in this species. Non-*Salmonella* Enterobacterales may be the common reservoir for *mcr*-1 in the DR, and a plasmid carrying the *mcr*-1 gene may pass sporadically into *Salmonella* in food items or even within the host patient. Although we mostly recovered *mcr* in *E. coli*, this investigation did not specifically target generic *E. coli*. The asymptomatic carriage of *mcr*-1 in travelers returning from the DR is more common than we recognize and could have implications for transmission to other bacteria in the U.S.

While it has been previously demonstrated that different plasmid types have been involved in the dissemination of *mcr*-1 genes globally, in this study, *mcr*-1 genes were consistently associated with an IncX4 plasmid replicon. A recent report similarly identified an association between *mcr*-1 genes and the IncX4 plasmid among clinical isolates (mainly *Salmonella* enterica) collected from U.S. patients reporting recent travel to the DR. In addition, the most common IncX4 contig arrangement reported here (Figure 4B) was also nearly identical to the plasmids found among the clinical isolates described (identity 99.9%, reference coverage 98.3% [32,732 of 33,292 bp]) [12]. Highly related *mcr*-1-bearing IncX4 plasmids have also been identified among Enterobacteriaceae in countries apart from the DR [25]. Given the diversity of plasmid types that have been reported to carry *mcr* genes, the persistent finding of the *mcr*-1 gene on IncX4 plasmids in the DR isolates is noteworthy. The observed similarity in plasmid type may result from a founder effect. Nonetheless, it is challenging to make epidemiological inferences based on the presence of plasmids alone.

Notably, most *mcr*-positive isolates (97.4%) from our study were MDR to three or more antibiotic classes. Previous research has demonstrated the coexistence of *mcr* genes with other genes that confer resistance to critically important antibiotics, including beta-lactams, macrolides, and quinolones [26,27]. Multidrug resistance to antibiotics recommended for managing human infections limits treatment options and poses a severe threat to public health. Most of the *E. coli* sequence types found in this study are not commonly related to human infections. However, investigations have demonstrated that *E. coli* ST10 infections occur in humans and animals—frequently linked to bovine surveillance—and such infections may contribute to the dissemination of the *mcr*-1 gene and IncX4 plasmid [28,29,30].

Discrepancies between the prevalence at the isolate and sample levels were observed across all animal sources. This discrepancy might be caused by many factors, such as the presence of non-viable (dead) bacteria that did not grow in the media, bacterial colony selection, or a substantial proportion of *mcr*-positive bacteria in the samples were types other than Enterobacteriaceae, and therefore not detected by the methods. Nonetheless, this study confirms the presence of *E. coli* carrying *mcr* genes, specifically *mcr*-1, in livestock and their environments in the DR. Official information about the use of antibiotics in food animal agriculture was not available in the DR; however, we observed that colistin was used in feed for swine during the study period.

Antimicrobial resistance is a complex ecological problem that affects human, animal, and environmental well-being and requires coordinated, multi-sector approaches such as One Health [31]. To curb the emergence and spread of antimicrobial resistance, actions are needed to promote the responsible use of antibiotics in human and veterinary medicine and to develop new ways to prevent and treat infections. The One Health approach provides healthcare professionals with a collaborative, multidisciplinary framework to address the complexity of AMR. Working together, healthcare experts can effectively reduce the impact of antimicrobial resistance on all aspects of health worldwide.

## 5. Conclusions

This study provides a unique example of a scenario where the emergence of a rare resistance gene among travelers led to investigating possible reservoirs for the gene in the travel destination. In this instance, evidence was available to demonstrate that *mcr*-1 was not already prevalent in food and animal reservoirs in the U.S., and that human infections harboring the gene were frequently associated with international travel to the DR. Our investigation identified the *mcr*-1 gene in *E. coli* from multiple food and animal reservoirs in the DR, particularly those related to swine, and further established that the same gene-plasmid combination (*mcr*-1/IncX4) found in both *E. coli* and *Salmonella* isolates from travelers was also found in *E. coli* from sources in the DR. These findings suggest potential agricultural reservoirs for these gene/plasmid combinations that might lead to colistin-resistant foodborne infections in humans. International collaborative efforts between stakeholders, including government, food animal industry partners, and research institutions, are needed to address and examine antibiotic use and improve antibiotic stewardship. Establishing a monitoring program using the One Health approach will provide data for policy decisions and guidance for animals and human health in a shared environment.

## Figures and Tables

**Figure 1 ijerph-20-05123-f001:**
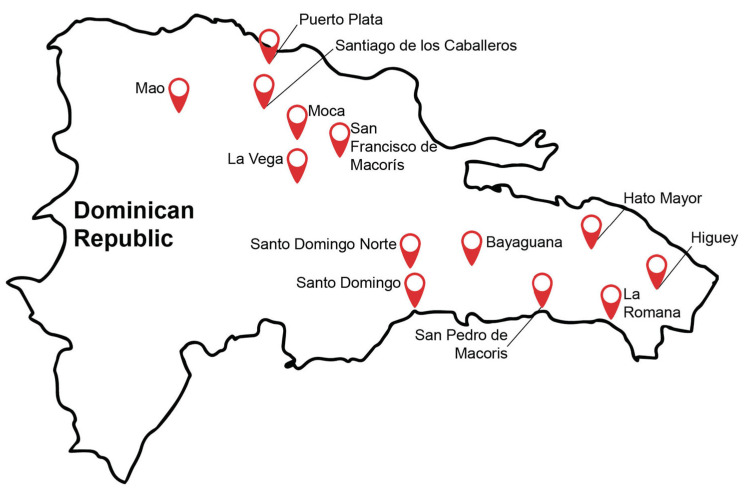
Sample collection sites in the Dominican Republic. Sampling locations throughout the country are marked with a red symbol.

**Figure 2 ijerph-20-05123-f002:**
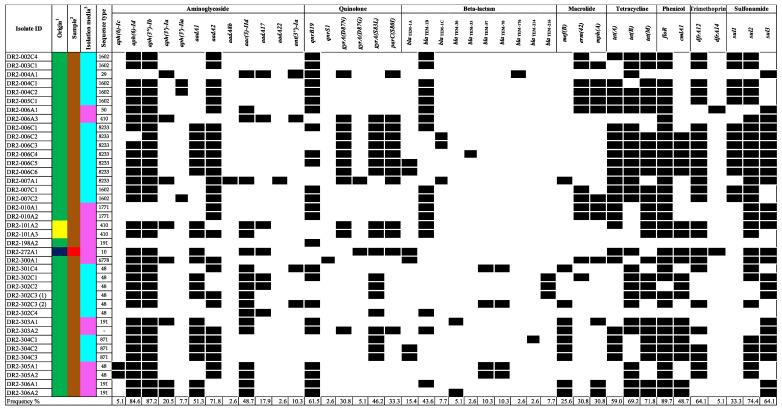
Distribution of AMR determinants on *mcr*-positive isolates for aminoglycosides, quinolones, beta-lactams, macrolides, tetracyclines, phenicol, trimethoprim, and sulfonamides. Black cells and white cells indicate the presence and absence of genes in the bacterial genome, respectively. For origin ^1^, the colors navy blue, yellow, and green, represent beef cattle, chicken, and swine, respectively. For sample ^2^, the colors brown and red represent feces and meat, respectively. For isolation media ^3^, the colors pink and light blue represent VRBG0 and XLT4/BGS, respectively.

**Figure 3 ijerph-20-05123-f003:**
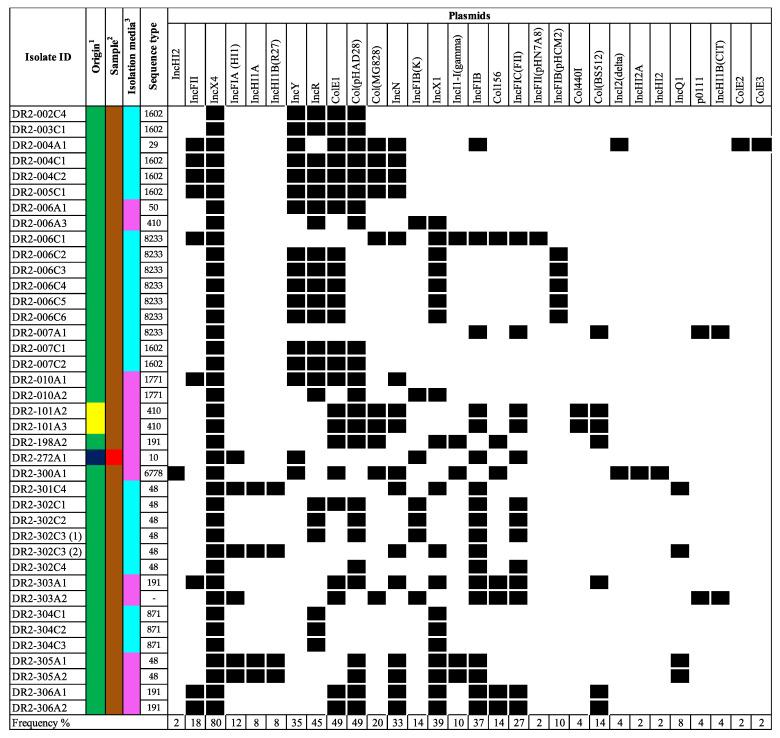
Black cells and white cells, indicate the presence or absence of plasmid replicon in the bacterial genome, respectively. For origin, the colors navy blue, yellow, and green, represent beef cattle, chicken, and swine, respectively. For origin ^1^, the colors navy blue, yellow, and green, represent beef cattle, chicken, and swine, respectively. For sample ^2^, the colors brown and red represent feces and meat, respectively. For isolation media ^3^, the colors pink and light blue represent VRBG0 and XLT4/BGS, respectively.

**Figure 4 ijerph-20-05123-f004:**
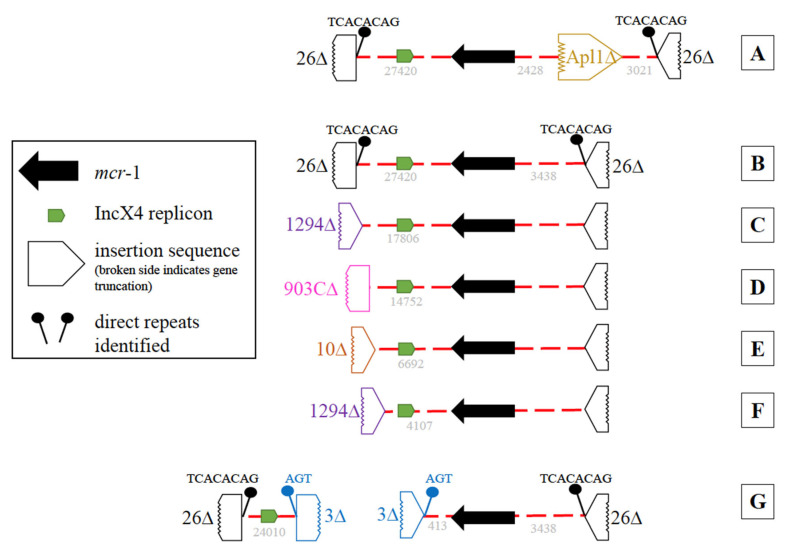
Diagrams of the genetic variation of the *mcr*-1 and IncX4 harboring contigs. An IncX4 plasmid replicon, *mcr*-1, and two partial IS26 with direct repeats (TCACACAG) were identified in variation (**A**–**G**). The IncX4 and *mcr*-1 were on the same contig in variations (**A**–**F**), which differed in length due to the presence of partial insertion sequences either at one end of the contig resulting in shorter contigs ((**B**–**F**); IS*1294*, IS*903C,* or IS*10*) or by the presence of a single IS*Apl1* ~2.4 kb upstream of the *mcr*-1 (**A**). In variation G, *mcr*-1 and IncX4 were on different contigs and portions of IS*3* with direct repeats (AGT) were found on one end of each contig.

**Table 1 ijerph-20-05123-t001:** Primers and probes for the *mcr*-1 to *mcr*-8 multiplex RT-PCR assay.

Primer/Probe	Sequence
* *mcr126*-F^1^	5′-GTAYTCTGTGCCGTGTATGTT-3′
* *mcr126*-R^2^	5′-TCCATCACGCCTTTTGAGTC-3′
*mcr3*-F^1^	5′-AAAACTGCACRGATGAAGAGC-3′
*mcr3*-R^2^	5′-TCATCTCTSCAATCACKAAATC-3′
*mcr4*-F^1^	5′-TTCAAAATTGCAGTCAAGAAGAAC-3′
*mcr4*-R^2^	5′-ACCACTTCACTGAGAATAAAATC-3′
*mcr5*-F^1^	5′-ACACGGCGCTGCTGTAC-3′
*mcr5*-R^2^	5′-TATGCCATGGAGATACAGGC-3′
*mcr7*-F^1^	5′-AACTGCAGCGATGAAGAGC-3′
*mcr7*-R^2^	5′-ATCATCTCTGCTATGACAAAATC-3′
*mcr8*-F^1^	5′-GATGCGTGACGTTGCTATG-3′
*mcr8*-R^2^	5′-TGTGCCATGAAGATATATTCCG-3′
*** mcr126*-P^3^	5′/56ROXN/TATGATGTCGATACCGCCAAATACCAAG/3IAbRQSp-3′
*** *mcr347*-P^3^	5′-/5HEX/AACACCTAYGACAACACYAT/3IABkFQ/-3′
**** *mcr58*-P^3^	5′-/56-FAM/TTTCYGATCATGGRGAATCGCT/3IABkFQ/-3′

^1^ Represents the forward primer sequence. ^2^ Represents the reverse primer sequence. ^3^ Refers to the probe design. * Primer (forward and reverse) detects *mcr*-1, *mcr*-2, and *mcr*-6 genes together. ** Probe detects *mcr*-1, *mcr*-2, and *mcr*-6 genes together. *** Probe detects *mcr*-3, *mcr*-4, and *mcr*-7 genes together. ******** Probe detects *mcr*-5, and *mcr*-8 genes together.

**Table 2 ijerph-20-05123-t002:** Proportion of RT-PCR presumptive *mcr*-positive samples by animal source and sample type.

Animal Source ^1^	Feces % (Positive/Total)	Feed % (Positive/Total)	Water % (Positive/Total)	Meat % (Positive/Total)	Total % (Positive/Total)
Beef cattle	52.5 (21/40)	0.0 (0/2)	60.0 (3/5)	65.2 (30/46)	58.1 (54/93)
Swine	97.4 (38/39)	46.2 (6/13)	50.0 (3/6)	66.7 (34/51)	74.3 (81/109)
Poultry	88.9 (32/36)	81.8 (9/11)	75.0 (3/4)	70.7 (41/58)	78.0 (85/109)
Total per sample type	79.1 (91/115)	57.7 (15/26)	60.0 (9/15)	67.7 (105/155)	70.7 (220/311)

^1^ Samples obtained from livestock farms, supermarkets, and butcher shops.

**Table 3 ijerph-20-05123-t003:** Proportion of RT-PCR presumptive *mcr*-positive isolates by animal source and sample type ^1^.

Animal Source	Feces % (Positive/Total)	Feed % (Positive/Total)	Water % (Positive/Total)	Meat % (Positive/Total)	Total per Animal Source % (Positive/Total)
Beef	4.2 (3/72)	0.0 (0/0)	0.0 (0/13)	0.9 (2/219)	1.6 (5/304)
Swine	14.7 (33/225)	18.8 (3/16)	0.0 (0/18)	0.0 (0/244)	7.2 (36/503)
Poultry	1.1 (2/187)	0.0 (0/47)	0.0 (0/7)	0.3 (1/306)	0.5 (3/547)
Total per Sample type	7.9 (38/484)	4.8 (3/63)	0.0 (0/38)	0.4 (3/769)	3.2 (44/1354)

^1^ Multiple isolates were collected from each sample; therefore, some strains may be represented twice.

## Data Availability

Sequences are available at NCBI (https://www.ncbi.nlm.nih.gov, accessed on 6 August 2022) under BioProject PRJNA761490.

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
