# Peer review of "First Known Report of mcr-Harboring Enterobacteriaceae in the Dominican Republic"

_ijerph, 2023, doi:10.3390/ijerph20065123_

Round 1

Reviewer 1 Report

The study is very interesting, the design is appropriate, as well as the methodology used. Here are some comments or observations:

1.      The study corresponds to a One Health approach and it would be convenient to expand the information in this regard, both in the introduction and in the discussion.

2.      Table 1, Primers and probes for the mcr-1 to mcr-8 multiplex RT-PCR assay, is unclear on the mcr126, mcr347, mcr-58 nomenclatures. It is suggested to review the correct nomenclature.

3.      The lack of correlation between RT-PCR and WGS regarding the detection of mcr1 genes is strange. To what do the authors attribute these findings?

4.      Table 4 would be better as supplementary information.

5.      In figure 2 and figure 3, it would be convenient to include the information on the sequence types, which would allow making inferences about the clonality of the isolates. These figures mention colors that do not exist.

6.      It would be convenient to analyze if there is a relationship between the Ecoli sequence types carrying mcr1 and the geographical regions in which they were isolated.

Author Response

General response:  authors appreciate the comments and recommendations provided, and we have addressed all of them.  A description of how the comments and recommendations were addressed are presented below; however, specific details can be seen in the new version of the manuscript uploaded, for which we used "track changes."

Comment 1. The study corresponds to a One Health approach and it would be convenient to expand the information in this regard, both in the introduction and in the discussion.

Response to comment 1. 

This comment has been addressed by incorporating information regarding the correlation between antimicrobial resistance and the One Health approach. The relevant information can be found in the sections from lines 31 to 40 and 1054 to 1061. 

Comment 2.  Table 1, Primers and probes for the mcr-1 to mcr-8 multiplex RT-PCR assay, is unclear on the mcr126, mcr347, mcr-58 nomenclatures. It is suggested to review the correct nomenclature.

Response to comment 1. The nomenclature we used are standard and have been used in other publications by other authors; however, we understand that the table perhaps was not very clear in explaining the abbreviations.  To improve and address this comment, we added a table caption below Table 1; corresponding numbers that link to the captions are presented in the table. The caption reads:

1Represents the forward primer sequence. 2 Represents the reverse primer sequence. 3Refers to the probe design. *Primer detects mcr1, mcr2, and mcr6 genes together. **Primer detects mcr3, mcr4, and mcr7 genes together. ***Primer detects mcr5, and mcr8 genes together.

Comment 3. The lack of correlation between RT-PCR and WGS regarding the detection of mcr1 genes is strange. To what do the authors attribute these findings?

Response to comment 3. The authors agree that it has not been easy to elucidate the reasons for such discrepancy.  The specificity of the molecular marker was evaluated in silico, and we did not find anything concerning.

One explanation could be the yield of recovery of the plasmid carrying the mcr1 gene during gDNA extraction. Indeed, PCR testing was performed on crude isolate lysis, while WGS was performed from a clean gDNA extraction. The difference of extraction method could have lead to a discrepancy between both tests. A second hypothesis is that the strains were transferred to new TSA plates or media multiple times to conduct various test, so it is possible that lost of the plasmid carrying the mcr gene occurred.  A third potential explanation is that for WGS we conducted short read sequence, which some times produces reads that are limited in length, producing incomplete gene coverage. Small genes such as mcr may not be fully covered.

Comment 4. Table 4 would be better as supplementary information.

Response to comment 4. Table 4 was moved to supplementary materials and labeled as Table S1.

Comment 5. In figure 2 and figure 3, it would be convenient to include the information on the sequence types, which would allow making inferences about the clonality of the isolates. These figures mention colors that do not exist.

Response to comment 5. The sequence type was incorporated into Figures 2 and 3. The caption of the figures was revised, and mention to colors was removed.

Comment 6. It would be convenient to analyze if there is a relationship between the Ecoli sequence types carrying mcr1 and the geographical regions in which they were isolated.

Response to comment 6. To address this recommendation, we have incorporated a column in Table S1 indicating the location where each sequence type was found in the country.

Reviewer 2 Report

Title: First known report of mcr-harboring Enterobacteriaceae in the Dominican Republic

Reference: ijerph-2209234

Type: Research

Overview:

The manuscript ijerph-2209234 entitled “First known report of mcr-harboring Enterobacteriaceae in the Dominican Republic” reports the identification of Escherichia coli carrying mcr-1.

General comments:

Overall, the manuscript is brief and well-written. However, minor changes are required.

The Abstract and Keywords sections should be reorganized.

The Introduction section is clear and sufficient.

The Material and Method section needs simplification.

The Result section needs simplification.

The Discussion section needs simplification.

The Conclusion section is clear and sufficient.

Specific comments:

Abstract: Please consider replacing with:

Abstract: Colistin is a last-resort antibiotic used to treat infections caused by multidrug-resistant Gram-negative bacteria. The CDC has reported that people traveling to the Dominican Republic become sick with colistin-resistance pathogens carrying the mcr-1 gene. Three hundred and eleven samples were collected, and 1,354 isolates were obtained. The overall real-time-PCR presumptive mcr prevalence was 70.7% (220/311) in samples and 3.2% (44/1,354) in isolates. All RT-PCR presumptive mcr-positive isolates (n=44) and a subset (n=133) of RT-PCR presumptive mcr-negative isolates were subjected to whole-genome sequencing. Based on the sequences, 39 isolates carried mcr genes (37 RT-PCR mcr-positive and two RT-PCR mcr-negative isolates), identified as Escherichia coli carrying mcr-1. The IncX4 plasmid replicon was identified in all isolates harboring mcr genes. Resistant determinants for other antibiotics important for human health were found in almost all isolates carrying mcr genes. This investigation aimed to identify mcr genes in Enterobacteriaceae isolated from food animal sources in the Dominican Republic.

Keywords: Please consider organizing by alphabetic order and removing the numbers.

antimicrobial resistance, colistin resistance, Enterobacteriaceae, Escherichia coli, IncX4, mcr genes

Introduction: Please consider replacing with:

Colistin, an old polymyxin, is a crucial last-resort antibiotic used to treat infections caused by multidrug-resistant (MDR) gram-negative bacteria. It is categorized as a critically important antibiotic in human health. The first plasmid-mediated colistin resistance mechanism in Enterobacteriaceae, mcr-1, was reported in China in 2015 [1]. Since then, retrospective and prospective screening efforts have globally detected mcr-1 in Enterobacteriaceae species collected from humans, animals, food, and the environment [2, 3]. Although mcr-1 is the most observed mcr gene, additional mcr genes (from mcr-2 to mcr-10) and their variations have been described [4, 5]. In the United States, where colistin use has never been approved for livestock, the first mcr gene reported was in an mcr-1-positive Escherichia coli isolate collected in 2016 from a urine sample from a woman in Pennsylvania. In the same year, the United States Department of Agriculture (USDA) reported mcr-1 in bacteria isolated from the intestinal tract of two pigs [6, 7]. Continued retrospective and prospective screening in the U.S. revealed that most mcr genes are rarely detected in routine surveillance of four major foodborne bacteria (SalmonellaCampylobacterEscherichia coli, and Enterococcus) from humans, animals at slaughter, and retail meats [8]. Such surveillance efforts have also linked mcr-positive clinical isolates and international travel within seven days before illness onset. From 2016 to 2021, 14 mcr-positive Salmonella were isolated from U.S. patients who traveled to the Dominican Republic (DR) [9, 10]. That data suggests that U.S. travelers may be exposed to and acquire enteric pathogens carrying mcr genes abroad by consuming contaminated food or water. This work aimed to identify mcr genes in Enterobacteriaceae isolated from animals, food, and animal environments from the DR to understand potential sources of human exposure better. The specific objectives were to: i) estimate sample-level prevalence by identifying mcr genes directly from enriched samples; ii) estimate isolate-level prevalence by detecting mcr genes in bacterial isolates obtained from the samples; iii) confirm taxonomic group, specific antimicrobial resistant determinants, and plasmid replicon types for RT-PCR presumptive mcr-positive isolates using whole genome sequencing (WGS).

Material and methods: Please consider replacing with:

2.1 Sample collection 

Samples were collected from raw meat (beef, pork, and broiler chickens[poultry]), feces from the corresponding food-producing animals, and water and feed for animal consumption. These samples were obtained from different country locations, such as Bayaguana, Hato Mayor, Higuey, La Romana, La Vega, Mao, Moca, Puerto Plata, San Francisco de Macoris, San Pedro de Macoris, Santiago de los Caballeros, Santo Domingo, and Santo Domingo Norte (Figure 1). Sampling sites and required authorizations to visit the farms were procured with the help of the Ministry of Agriculture in the DR. 

Meat samples were purchased from supermarkets and small butcher shops, placed into sterile bags, and transported the same day under refrigeration for further testing. 

Fecal samples from beef cattle and swine consisted of rectal specimens collected by government veterinarians using sterile palpation gloves and placed into sterile cups. 

Poultry litter was sampled using sterile pre-moistened boot swabs (Envirobootie™, Hardy Diagnostics). Two individuals wore boot swabs and walked around the poultry pen for 5 min. Each boot swab (left and right foot) was placed separately into a sterile bag and treated as an independent sample.

Water samples were obtained from animal waterers from farming operation facilities (beef cattle, swine, or poultry) and placed into sterile cups.

Animal feed was obtained either from open containers at the farms or warehouses supplying pig farms and poultry houses. Beef cattle feed samples were limited to only one farm. 

All samples were labeled accordingly (date, type of sample, animal origin, location, and name of farm/market/shop) and transported the same day in coolers to the official testing microbiology laboratory affiliated with the Ministry of Agriculture in Santo Domingo.

2.2 Sample preparation and enrichment

Buffered Peptone Water was supplemented with 1 mg/L colistin (BPW1) and BPW without colistin (BPW0) were prepared following the manufacturer's instructions. Samples were separately added to 90 mL of each one of the enrichment solutions (BPW0 and BPW1). Meat cuts and boot swabs were rinsed with 250 mL of BPW, and 10 mL of the rinse was transferred into each enrichment solution. For fecal and feed samples, 10 g were used for enrichment. For water samples, 10 mL were transferred to each enrichment bottle. Samples were incubated at 37°C for 24h.

2.3 Enterobacteriaceae detection and isolation

After incubation, samples were cultured on Violet Red Bile Glucose Agar plates (VRBG, BD Difco®, Sparks, MD, United States) supplemented with colistin (1 mg/L [VRBG1] and 0 mg/L [VRBG0]) and incubated at 37ºC for 24h.

Presumptive Enterobacteriaceae colonies, selected based on typical morphology, were cultured on Tryptic Soy Agar (TSA, MilliporeSigma, MA 01821, United States), supplemented with colistin (1 mg/L [TSA1] and 0 mg/L [TSA0]), and incubated at 37°C for 24h. Colonies were transferred to TSA slants supplemented with colistin and stored at 4°C for analysis. 

2.4 Salmonella detection and isolation

Samples were pre-enriched in BPW and incubated at 37°C for 24 h. A subsequent selective enrichment was conducted by transferring 1 mL into 9 mL of Rappaport-Vassiliadis broth (RV, BD Difco®, Sparks, MD, United States) and 1 mL into 9 mL of Tetrathionate broth (TT, BD Difco®, Sparks, MD, United States), followed by incubation at 42°C for 24 h. After incubation, each selective enrichment, was transferred to Xylose Lysine Tergitol 4 agar plates (XLT4, BD Difco®, Sparks, MD, United States) and Brilliant Green Sulfa Agar (BGS, BD Difco®, Sparks, MD, United States), and incubated at 37°C for 24 h. Salmonella was isolated following steps similar to Enterobacteriaceae, including culture on TSA and transferring to new TSA slants.

2.5 DNA extraction from samples

DNA was extracted from the BPW enrichments not supplemented with colistin. QIAamp PowerFecal DNA Kit (QIAGEN® Group, Germantown, MD, United States) was used for fecal samples, and the InstaGene matrix kit (Bio-Rad Laboratories, Hercules, California) for feed, water, and meat cut samples. Manufacturers' protocols were followed in both cases.

2.6 DNA extraction from isolates

Presumptive Salmonella isolates were transferred to new TSA plates without colistin and incubated overnight at 37°C. Colonies were selected and inoculated in Luria broth (LB, Remel, Lenexa, KS, United States) and incubated at 37°C for 18h in an incubator shaker. The presumptive Enterobacteriaceae isolates were transferred to VRBG supplemented with the colistin and incubated at 37°C for 24 h. After incubation, one colony was inoculated in TSB supplemented with the corresponding colistin and incubated at 37°C for 18 h in an incubator shaker. An aliquot of 150 ml was used to extract genomic DNA using a boiling method. Samples were centrifuged at 4,500 rpm for 3 min at 4°C, and the pellet was resuspended in 30 ml of molecular-grade water and centrifuged again. Samples were stored at -20ºC until further analysis. Separately, 850 µl of the overnight TSB enrichment was preserved with 150 µl of 80% glycerol at -80°C for in-house culture collection.

2.7 Multiplex RT-PCR for mcr gene detection 

Multiplex real-time polymerase chain reaction (RT-PCR) was conducted to detect mcr-1 to mcr-8 genes. The multiplex RT-PCR assay was designed based on reference sequences accessible in the ResFinder database. It included 14 variants of mcr1, two variants of mcr2, 25 variants of mcr3, six variants of mcr4, two variants of mcr5, one variant of mcr6, one variant of mcr7 and one variant of mcr8. The multiplex RT-PCR was optimized as follows: 25 ml PCR reactions including 1x Brilliant II QPCR Master Mix (Agilent, Santa Clara, CA), mcr-3, mcr-4, mcr-5, mcr-7, and mcr-8 forward and reverse primers at the final concentration of 0.6µM, mcr126 forward and reverse primers at 0.3µM, mcr126 probe at 0.7µM, and mcr58 and mcr347 probes at 0.8µM and a DNA input of 3 ml. The RT-PCR thermal program was set up as follows: an initial denaturation of 10 min at 95°C, followed by 35 cycles including denaturation for 30 sec at 95°C and an annealing/elongation step of 60 sec at 66°C. Primers and probes were designed and labelled, as shown in Table 1.

Positive controls for mcr-1 and mcr-5 were obtained using Salmonella reference strains 12CEB2196SAL [12] and S12LNR3592 [13] (Anses, Maisons-Alfort, France). Positive controls for mcr-2, mcr-3, mcr-4, mcr-6, mcr-7, and mcr-8 were obtained by electroporating Salmonella 10TTU468x (Texas Tech University Food Microbiology Laboratory culture collection) with AmpR plasmids, cloned by the manufacturer (IDT, Newark, NJ) with mcr-2.1, mcr-3.1, mcr-4.1, mcr-6.1, mcr-7.1 or mcr-8.1 sequences obtained from ResFinder database. The mcr-9 and mcr-10 genes were not included since they were not described when this study was designed.

2.8 Whole Genome Sequencing 

All RT-PCR presumptive mcr-positive isolates were subjected to whole genome sequencing (WGS). For sequencing, a subset of RT-PCR presumptive mcr-negative isolates (n=133) was randomly selected (using the RAND Microsoft Excel function with weight). DNA extraction was performed using GenEluteTM bacterial genomic DNA kit (Sigma-Aldrich, NA2100, NA2110, or NA2120, St Louis, MO) following the manufacturer protocol. Libraries were constructed with 5 µl genomic DNA (~100-200 ng ) and Nextera DNA Flex Library Prep Kit (Illumina) and quantified on the Fluorometer using PicoGreen, according to the manufacturer's protocol. The pool was analyzed using an Illumina MiSeq Reagent Nano kit v2 (300 cycles) on Illumina MiSeq. Then, 10 µl of the pooled library at a final concentration of 200 pM were sequenced using an Illumina 250 × 250 NovaSeq SP Flow Cell (500 cycles) on the Illumina NovaSeq-6000 sequencing facility (TTU Center for Genetics and Biotechnology, Lubbock, TX, United States). Genotypic characterization was performed on all 44 RT-PCR presumptive mcr-positive isolates. From the randomly selected 133 RT-PCR presumptive mcr-negative isolates, 20 were discarded due to low coverage during the short read sequencing. Therefore, plasmid characterization and AMR profile were conducted on 113 of them. Genus and species identification was performed using FastANI (https://github.com/ParBLiSS/FastANI) and the Genome Taxonomy Database (https://data.ace.uq.edu.au/public/gtdb/data/releases/release95/). De novo assemblies were generated using shovill v1.0.9 (https://github.com/tseemann/shovill), and contigs with coverage below 10% average genome coverage were excluded. Staramr v0.4.0 (https://github.com/phac-nml/staramr) was used to screen assemblies for resistance determinants using the ResFinder database (Center for Genomic Epidemiology [CGE], https://cge.cbs.dtu.dk; downloaded 30JUL2020, 90% identity, 50% cutoff) and CGE PointFinder scheme for Salmonella spp. ARIBA v2.12.0 and the CGE PointFinder database were used to screen E. coli for point mutations. Plasmid replicons were identified using abricate v0.8.10 and a database adapted from CGE PlasmidFinder (90% identity, 60% cutoff). Multilocus sequence type was determined from de novo assemblies using the Tseemann MLST tool (https://github.com/tseemann/mlst) and the PubMLST database [14]

Results: Please consider replacing with:

In total, 311 samples were collected from 17 farms, 21 supermarkets, and 12 butcher shops in the DR. Table 2 shows the detail of each sample collected and the proportion of RT-PCR presumptive mcr-positive samples by animal source and sample type. Based on the multiplex RT-PCR, 70.7% (n=220) of the samples collected carried one or more mcr genes. The proportion of RT-PCR presumptive mcr-positive samples collected from beef cattle, swine, and poultry was 58.1% (54/93), 74.3% (81/109), and 78.0% (85/109), respectively. The overall proportion of RT-PCR presumptive mcr-positive samples based on sample type for feces, feed, water, and meat, was 79.1% (91/115), 57.7% (15/26), 60.0% (9/15), 67.7% (105/155), respectively.

3.1 mcr prevalence at the isolate level

A total of 1,354 isolates were recovered from 311 samples. Multiplex RT-PCR showed that 3.2% (n=44) of the isolates carried mcr genes. Within the 44 presumptive mcr-positive isolates, 81.8% (n=36) were obtained from swine sources, 11.4% (n=5) from beef cattle and 6.8% (n=3) from poultry (Table 3). When grouping isolates by sample type, the highest mcr proportion was observed among isolates recovered from fecal samples at (7.9%; n=38), feed (4.8%; n=3), and meat samples (0.4%; n=3). 

3.2 Whole genome sequencing analysis for mcr confirmation

Presumptive RT-PCR mcr-positive (n=44) and mcr-negative (n=133) isolates were subjected to WGS analysis. Of the 44 RT-PCR presumptive mcr-positive isolates, 84.1% (n=37) were confirmed as mcr-positive (Table 4). WGS also revealed that two RT-PCR presumptive mcr-negative isolates carried mcr genes, and both were identified as E. coli (mcr-1-positive E. coli [DR2-004A1 and DR2-007A1]; Table 4). Altogether, the 39 mcr-1-positive E. coli isolates included the following sequence types: ST10, ST29, ST48, ST50, ST191, ST410, ST871, ST1602, ST1771, ST6778, and ST8233. From the 133 mcr-negative randomly selected isolates subjected to WGS, 113 sequences were included in our analysis (20 sequences did not have the appropriate coverage). Sequences were classified as Morganella morganii (30.1%; n=40), Salmonella enterica (24.1%; n=32), E. coli (8.3%; n=11), Proteus spp (6.8%; n=9), Providencia spp (5.3%; n=7), Hafnia spp (3.4%; n=5), Citrobacter spp (3.0%; n=4), Enterobacter spp (1.5% (n=2), Shewanella algae (1.5%; n=2), and Moellerella wisconsensis (0.7%; n=1). Of the 39 mcr-positive isolates, 92.3% (n=36) originated from swine feces, 5.1% (n=2) from poultry feed, and 2.6% (n=1) from beef (meat) (Table 4). These 39 mcr-positive E. coli isolates were recovered from 17 out of the 311 samples. Furthermore, these isolates were obtained from samples collected in either the Santo Domingo Province (Santo Domingo and North Santo Domingo; n=8) or North-Central DR (La Vega [n=7] and Moca [n=2]). 

The mcr-positive isolates harbored many additional antimicrobial resistance determinants to aminoglycosides, quinolones, beta-lactams, macrolides, tetracyclines, phenicol, and/or folate pathway inhibitors (including trimethoprim and sulfonamides (Figure 2). From the 39 isolates carrying mcr genes, 38 harbored antimicrobial-resistant genes to three or more antimicrobial classes. Moreover, the 38 isolates had resistance determinants for additional critically important antimicrobials for human medicine. These include aminoglycosides, quinolones, macrolides, and subclasses of the beta-lactam class. The aminoglycoside modifying enzymes aph(3”)-Ib (87.2%) and aph(6)-Id (84.6%) were commonly present. The quinolone resistance determining region (QRDR) mutation in gyrA(S83L) was present in 51.3% (n=20), gyrA(D87N) in 28.2% (n=11), gyrA(D87G) in 2.6% (n=1), and parC(S80I) in 33.3% (n=13) of the isolates. Moreover, the qnrB19 and qnrS1 genes, which confer reduced susceptibility to quinolones in Enterobacteriaceae, were observed in 61.5% (n=24) and 2.6% (n=1), respectively. The blaTEM gene variants, especially blaTEM-1B, were present in 43.6% (n=17). Furthermore, the mef(B) macrolide-efflux pump gene was found in 25.6% (n=10) of the isolates, followed by the presence of both mph(A) and erm(42) genes in 30.8% (n=12). Among the mcr-positive isolates, the most predominant plasmid replicons found were IncX4 (100.0%), ColE1 (61.5%), Col(pHAD28) (61.5%), and IncR (56.4%) (Figure 3). Other replicon types known for carrying mcr genes were less common among the mcr-positive isolates and included IncX1 (48.7%), IncFIB (46.2%), IncFII (23.1%), and IncHI2 (2.6%). As previously noted, all 39 mcr-1-positive isolates harbored an IncX4 plasmid replicon. In most mcr-1-positive isolates (35/39), mcr-1 and IncX4 were on the same contig. However, six different contig variations were observed (Figure 4A-F). The most common (Figure 4B, n=23harbored a partial IS26 with direct repeats (TCACACAG) at each end of the contig. Contig variations C-F (Figure 4) also contained mcr-1 and IncX4 on the same contig; however, they differed by the insertion sequences (IS1294, IS903C, or IS10) at one end resulting in shorter contigs. When mcr-1 and the IncX4 replicon were found on different contigs (Figure 4G, n=4), portions of IS3 with direct repeats (AGT) were found on one end of each contig, indicating these contigs may be linked (Figure 4G). In three of the 39, a single partial (1,023/1,070 bp) insertion sequence, ISApl1, was found 2,428 bp upstream of the mcr-1 (Figure 4A). No evidence of an ISApl1 was identified adjacent to mcr-1 in the remaining 36 isolates.

Discussion: Please consider replacing with:

The dissemination of MDR Gram-negative bacteria carrying genes that confer resistance to critically important antibiotics through the food chain represents a global public health threat. Mobile colistin resistance genes have been detected in Gram-negative bacteria around the world. In many geographic regions, the prevalence of mcr genes remains undiscovered due to the lack of systematic research on this topic. Strengthening knowledge related to the prevalence and distribution patterns of resistance genes, such as mcr, across pathogens and geographical regions can be foundational to inform action against their selection and dissemination. The U.S. has national surveillance efforts in place, and it is known that mcr genes are rare in the country. Thus, it has been observed that many U.S. patients with mcr-1 genes were travelers to the DR [10]. The mobility of humans through traveling around the world is also a critical factor in disseminating antibiotic-resistant organisms. Travelers are exposed to bacteria through food or water and could subsequently acquire and spread mcr-positive Enterobacteriaceae. This study is the first known report of the crude prevalence of mcr genes using RT-PCR at the sample, and bacterial isolate levels obtained from food-producing animals and environments in the DR. Whole genome sequencing was used to confirm the presence of mcr genes in the recovered E. coli isolates. 

Our RT-PCR results indicate that mcr genes were commonly found in samples from beef cattle, swine, poultry, and environments in the DR. A higher prevalence of mcr genes was observed in swine compared to beef cattle and poultryWhole genome sequencing data also indicated that mcr-1 was common in E. coli from swine in the DR, particularly in the feces. Other researchers have reported plasmid-mediated colistin-resistant genes in swine from other geographic regions. For example, a study in China found that the prevalence of mcr genes at the sample level was higher in pigs than in other animals, such as poultry. The high level was associated with the prolonged and widespread use of colistin as a growth promoter in animals in China [15, 16, 17]. Another study on pig farms in Portugal reported a high prevalence of the mcr-1 gene in E. coli isolates [18]. Researchers have attributed the frequency of mcr genes in swine to the use of colistin sulfate for growth promotion or to control infections caused by E. coli in pigs, such as post-weaning diarrhea and edema disease [1, 19, 20]. In the DR, 36 colistin-containing products are approved for veterinary use in poultry, cattle, goats, sheep, pigs, cats, or dogs by the Ministry of Agriculture [21]. However, there are no known official animal antibiotic regulations, and antibiotic use and sales data still need to be included. During the sample collection of this project, we observed farmers adding colistin to the animal feed on swine farms but not on other animal holdings.

Interestingly, only one mcr gene variant (mcr-1) was found in this study - in one organism, E. coli - despite using targeted isolation methods to identify eight variants of mcr genes among Enterobacteriaceae more generally. Despite reports from many countries of Salmonella spp. in food, animals, and clinical specimens carrying mcr genes, and reports of mcr-positive Salmonella spp. isolated from U.S. travelers returning from the DR [10], mcr genes were not detected in this speciesNon-Salmonella Enterobacterales may be the common reservoir for mcr-1 in the DR. A plasmid carrying the mcr-1 gene may pass sporadically into Salmonella in food items or even within the host patient. Although we mostly recovered mcr in E. coli, this investigation did not specifically target generic E. coli. The asymptomatic carriage of mcr-1 in travelers returning from the DR is more common than we recognize and could have implications for transmission to other bacteria in the U.S. 

While it has been previously demonstrated that different plasmid types have been involved in the dissemination of mcr-1 genes globally, in this study, mcr-1 genes were consistently associated with an IncX4 plasmid replicon. A recent report similarly identified an association between mcr-1 genes and the IncX4 plasmid among clinical isolates (mainly Salmonella enterica) collected from U.S. patients reporting recent travel to the DR. In addition, the most common IncX4 contig arrangement reported here (Figure 5B) was also nearly identical to the plasmids found among the clinical isolates described (identity 99.9%, reference coverage 98.3% [32,7324 of 33,292 bp]) [10]. Highly related mcr-1-bearing IncX4 plasmids have also been identified among Enterobacteriaceae in countries apart from the DR [22]. Given the diversity of plasmid types that have been reported to carry mcr genes, the persistent finding of the mcr-1 gene on IncX4 plasmids in DR isolates is noteworthy. The observed similarity in plasmid type may result from a founder effect.

Nonetheless, it is challenging to make epidemiological inferences based on the presence of plasmids alone. Notably, most mcr-positive isolates (97.4%) were MDR to three or more antibiotic classes. Previous research has demonstrated the coexistence of mcr genes with other genes that confer resistance to critically important antibiotics, including beta-lactams, macrolides, and quinolones [23-25]. Multidrug resistance to antibiotics recommended for managing human infections limits treatment options and poses a severe threat to public health. Most of the E. coli sequence types found in this study are not commonly related to human infections. However, investigations have demonstrated that E. coli ST10 infections occur in humans and animals - frequently linked to bovine surveillance - and such infections may contribute to the dissemination of the mcr-1 gene and the IncX4 plasmid [26-28].

Discrepancies between the prevalence at the isolate and sample levels were observed across all animal sources. This discrepancy might be caused by many factors, such as the presence of non-viable (dead) bacteria that did not grow in the media, bacterial colony selection, or a substantial proportion of mcr-positive bacteria in the samples were types other than Enterobacteriaceae, and therefore not detected by the methods. Nonetheless, this study confirms the presence of E. coli carrying mcr genes, specifically mcr-1, in livestock and their environments in the DR. Official information about the use of antibiotics in food animal agriculture was not available in the DR; however, we did observe that colistin was used in feeds for swine during the study period.

Conclusions: Please consider replacing with:

This study provides a unique example of a scenario where the emergence of a rare resistance gene among travelers led to investigating possible reservoirs for the gene in the travel destination. In this instance, evidence was available to demonstrate that mcr-1 was not already prevalent in food and animal reservoirs in the United States, and that human infections harboring the gene were frequently associated with international travel to the DR. Our investigation identified the mcr-1 gene in E. coli from multiple food and animal reservoirs in the DR, particularly those related to swine, and further established that the same gene-plasmid combination (mcr-1/IncX4) found in both E. coli and Salmonella isolates from travelers was also found in E. coli from sources in the DR. These findings suggest potential agricultural reservoirs for these gene/plasmid combinations that might lead to colistin-resistant foodborne infections in humans. International collaborative efforts between stakeholders, including government, food animal industry partners, and research institutions, are needed to address and examine antibiotic use and improve antibiotic stewardship. Establishing a monitoring program using the One Health approach will provide data for policy decisions and guidance regarding the health of animals and humans in a shared environment.

Scientific comments:

Lines 21-22: The overall real-time-PCR presumptive mcr prevalence was 70.7% (220/311) in the samples and 3.2% (44/1,354) in the isolates. Please explain this sentence.

Lines 24-25: Based on the sequences, 39 isolates carried mcr genes (37 RT-PCR mcr-positive and two RT-PCR mcr-negative isolates), identified as Escherichia coli carrying mcr-1. Please explain this sentence.

Author Response

General Response to Reviewer 2:  

We have read the entire document and the recommendations made to our writing. Reviewer 2 made some suggestions directly to the manuscript, and most of them were accepted as we all considered that they improved the document.  Some deletions made by the reviewer were not accepted as the authors agreed that the text was needed.  Please see the new version of the manuscript, in which the track changes feature was used.  Below we describe how we addressed some specific comments:

Specific clarifications:

Reviewer comment: Lines 21-22: The overall real-time-PCR presumptive mcr prevalence was 70.7% (220/311) in the samples and 3.2% (44/1,354) in the isolates. Please explain this sentence.

Autors' response: The comment was addressed by rephrasing the sentence. It can be found in lines 17-19.

Reviewer comment: Lines 24-25: Based on the sequences, 39 isolates carried mcr genes(37 RT-PCR mcr-positive and two RT-PCR mcr-negative isolates), identified as Escherichia coli carrying mcr-1. Please explain this sentence.

Authors' response: The comment was addressed by rephrasing the sentence. It can be found in lines 20-23.

Reviewer comment: The Abstract and Keywords sections should be reorganized.

Authors' response: The abstract and keywords were organized as suggested.

Reviewer comment: The Introduction section is clear and sufficient.

Authors' response: Modifications were made as suggested. 

Reviewer comment: The Material and Method section needs simplification.

Authors' response: Modifications were made as suggested. 

Reviewer comment: The Result section needs simplification.

Authors' response: Modifications were made as suggested, but keeping some of the original text.

Reviewer comment: The Discussion section needs simplification.

Authors' response: Modifications were made as suggested, but keeping some of the original text.

Reviewer comment: The Conclusion section is clear and sufficient.

Authors' response: Modifications were made as suggested, but keeping some of the original text.

Reviewer 3 Report

Dear Editor,

Thank you for the opportunity to review this manuscript, “Title: First known report of mcr-harboring Enterobacteriaceae in the Dominican Republic”.  After reading the manuscript, I thought that the study was interesting.  The work might be appropriate for publishing after taking into account the suggestions and comments below.

General Comments:

Authors investigated to identify mcr genes in Enterobacteriaeae isolated from animals, food and animal environments in the Dominican Republic to better understand potential sources of human exposure.  Authors demonstrated that mcr-1 was not already prevalent in food and animal reservoirs in the United States, and that human infections harboring the gene were frequently associated with international travel to the Dominican Republic.  The findings in this study suggest possible agricultural reservoirs for these gene/plasmid combinations that might lead to colistin-resistant foodborne infections in humans.

This research could be of interest for the field. English language to be improved in the manuscript.

Author Response

The authors appreciate your time to review our manuscript and provide input. As per your recommendation, we did review the entire document, and some improvements in writing were made.  The document, in general has some improvements in terms of the English language.